# Alberta Childhood Cancer Survivorship Research Program

**DOI:** 10.3390/cancers15153932

**Published:** 2023-08-02

**Authors:** Andrew Harper, Fiona Schulte, Gregory M. T. Guilcher, Tony H. Truong, Kathleen Reynolds, Maria Spavor, Natalie Logie, Joon Lee, Miranda M. Fidler-Benaoudia

**Affiliations:** 1Department of Cancer Epidemiology and Prevention Research, Cancer Care Alberta, Alberta Health Services, Calgary, AB T2S 3C3, Canada; andrew.harper@ahs.ca; 2Department of Oncology, Cumming School of Medicine, University of Calgary, Calgary, AB T2N 1N4, Canada; fsmschul@ucalgary.ca (F.S.); greg.guilcher@ahs.ca (G.M.T.G.); tony.truong@ahs.ca (T.H.T.); 3Long Term Survivor’s Clinic, Alberta Children’s Hospital, Calgary, AB T2N 4N1, Canada; kathya.reynolds@ahs.ca; 4Department of Pediatrics, Cumming School of Medicine, University of Calgary, Calgary, AB T2N 4N1, Canada; 5Department of Medicine, Faculty of Family Medicine, University of Calgary, Calgary, AB T2N 4N1, Canada; 6Department of Pediatrics, Faculty of Medicine and Dentistry, University of Alberta, Edmonton, AB T6G 2R7, Canada; maria.spavor@ahs.ca; 7Division of Radiation Oncology, Tom Baker Cancer Centre, Alberta Health Services, Calgary, AB T2N 4N2, Canada; natalie.logie@ahs.ca; 8Data Intelligence for Health Lab, Cumming School of Medicine, University of Calgary, Calgary, AB T2N 4N1, Canada; joonwu.lee@ucalgary.ca; 9Department of Cardiac Sciences, Cumming School of Medicine, University of Calgary, Calgary, AB T2N 4N1, Canada; 10Department of Community Health Sciences, Cumming School of Medicine, University of Calgary, Calgary, AB T2N 4N1, Canada

**Keywords:** childhood cancer, pediatric cancer, survivorship, epidemiology

## Abstract

**Simple Summary:**

Treatments used to cure childhood cancer can have negative long-term impacts on physical health and well-being. Here, we present the Alberta Childhood Cancer Survivorship Research Program, its foundational cohort, and descriptive statistics of outcomes ascertained through data linkage. To this end, 2581 survivors of childhood cancer were included in the cohort, the majority of which were male, diagnosed between the ages of 0 and 4 years, with leukemia, central nervous system tumor, or lymphoma. By the study exit date, the median time since diagnosis was 5.6 years overall and 10.3 years for 5-year survivors. During the follow-up time, 94 subsequent cancers were diagnosed, 16,669 inpatient and 445,150 ambulatory/outpatient events occurred, 396,074 claims were reported, and 408 survivors died. The results from this research program seek to inform and improve clinical care and reduce cancer-related sequelae.

**Abstract:**

Adverse outcomes after childhood cancer have been assessed in a range of settings, but most existing studies are historical and ascertain outcomes only after 5-year survival. Here, we describe the Alberta Childhood Cancer Survivorship Research Program and its foundational retrospective, population-based cohort of Albertan residents diagnosed with a first primary neoplasm between the ages of 0 and 17 years from 1 January 2001 to 31 December 2018. The cohort was established in collaboration with the Alberta Cancer Registry and Cancer in Young People in Canada program and has been linked to existing administrative health databases and patient-reported outcome questionnaires. The cohort comprised 2581 survivors of childhood cancer, 1385 (53.7%) of whom were 5-year survivors. Approximately 48% of the cohort was female, 46% of the cohort was diagnosed between 0 and 4 years of age, and the most frequent diagnoses were leukemias (25.3%), central nervous system tumors (24.2%), and lymphomas (14.9%). Detailed treatment information was available for 1745 survivors (67.6%), with manual abstraction ongoing for those with missing data. By the study exit date, the median time since diagnosis was 5.6 years overall and 10.3 years for 5-year survivors. During the follow-up time, 94 subsequent primary cancers were diagnosed, 16,669 inpatient and 445,150 ambulatory/outpatient events occurred, 396,074 claims were reported, and 408 survivors died. The results from this research program seek to inform and improve clinical care and reduce cancer-related sequelae via tertiary prevention strategies.

## 1. Introduction

Survivorship from childhood cancer has increased substantially since the 1970s due to improvements in treatment and supportive care. Currently, approximately 80% of children diagnosed with a malignancy in Westernized countries will become 5-year survivors [1]. Although this is encouraging, these individuals continue to experience significant sequelae due to the disease itself and its treatment, with studies reporting increased risks of premature mortality [2,3,4,5], subsequent primary neoplasms (SPNs) [6,7,8], chronic health conditions [9,10,11], poor quality of life [11,12,13], and many other adverse outcomes [2,14,15,16,17]. While these outcomes have been assessed in a range of settings, most studies to date are historical (i.e., diagnoses before the year 2000) and ascertain adverse outcomes beginning only after 5-year survival [18,19]. Thus, the true impact of cancer and its treatment in contemporary childhood cancer survivor populations is largely unknown and warrants investigation given the dynamic nature of cancer therapies.

Here, we report on the study setting, design, objectives, and methodology of the Alberta Childhood Cancer Survivorship Research Program. The goal of the research program is to reduce the lifelong impact of childhood cancer. Current objectives include (i) quantifying the burden of physical and psychosocial late effects, (ii) identifying risk factors and examining dose–response relationships between anti-neoplastic therapies and adverse outcomes, and (iii) determining the impact of long-term survivorship clinics (LTSCs) at mitigating and preventing late effects. The expectation is that the program will grow to include biological samples and basic science initiatives and explore the association of newer immunotherapies and proton therapy with short and late effects in the future. The results generated from the program will potentially inform tertiary preventive strategies and follow-up care models in Alberta and beyond.

## 2. Materials and Methods

### 2.1. Study Setting

With an area of 661,688 km^2^, Alberta has a population of approximately 4 million people, about one million (25.1%) of whom are between the ages of 0 and 17 years [20]. Alberta has a publicly funded one-payer universal healthcare system administered by Alberta Health Services (AHS) [21]. Cancer care for children is provided primarily at the Alberta Children’s Hospital in Calgary and Stollery Children’s Hospital in Edmonton, although some children will be treated at tertiary, regional, or community centers distributed across the province. Additionally, many hospitals provide limited care to cancer patients, primarily surgery [21], chemotherapy, and pain relief [22].

### 2.2. Study Population

The Alberta Childhood Cancer Survivorship Research Program utilizes a retrospective, population-based cohort of Albertan residents diagnosed with a first primary neoplasm between the ages of 0 and 17 years from 1 January 2001 to 31 December 2018 as its foundation. The cohort was established in collaboration with the Alberta Cancer Registry (ACR) [21] and Cancer in Young People in Canada (CYP-C) program [23]. The ACR provided information on the date of birth, sex, zone of residence, date of diagnosis, tumor morphology and topography, American Joint Committee on Cancer stage, and basic information on initial treatment (i.e., presence or absence of surgery, chemotherapy, radiotherapy, and hormone therapy). Each cancer was classified according to the International Classification of Childhood Cancer, 3rd edition [24]. The ACR is operated by Cancer Care Alberta within AHS and is mandated by the Regional Health Authorities Act of Alberta, and thus the ascertainment of all cancer diagnoses is considered complete within two years [25]. The ACR has achieved Gold Certification from the North American Association of Central Cancer Registries, which is based on the completeness of data, timely reporting, and other measures that judge data quality [26].

As anti-neoplastic therapies are the main risk factors for late effects among survivors of childhood cancer, the ACR childhood cancer cohort was linked to the Alberta Children’s Hospital and Stollery Children’s Hospital institutional CYP-C databases. The CYP-C program was launched in 2009 to actively follow children who were diagnosed with cancer [24] and treated at 1 of the 17 pediatric oncology centers across Canada. The cohort includes all cases diagnosed before the age of 18 years from 2001 onwards and systematically collects data on demographics, diagnostic details, location and timing of care, and detailed treatment information [23].

### 2.3. Long-Term Survivor Clinic (LTSC)

The Stollery Children’s Hospital and Alberta Children’s Hospital provide all acute pediatric cancer treatment and late effects surveillance in Alberta, Canada [27]. The Northern Alberta Childhood Cancer Survivor Program at the Stollery Children’s Hospital serves survivors of childhood cancer from northern Alberta, Northwest Territories, and Nunavut, and follows survivors at risk for late effects from two years off therapy into and throughout adulthood, with no age limit. It is served by a multidisciplinary team led by a dedicated pediatric oncologist with expertise in late effects. The Alberta Children’s Hospital LTSC serves survivors from southern Alberta and is staffed with a dedicated family physician and nurse practitioner trained in late effects care. Although the Alberta Children’s Hospital LTSC currently utilizes the same practice model as the Stollery Children’s Hospital, it previously only followed survivors from 2 years off therapy until age 20 or 10 years off treatment (whichever came last), at which point the adult survivors were discharged to community primary care providers with treatment summaries and survivor care plans.

### 2.4. Late-Effect Ascertainment

#### 2.4.1. Alberta Cancer Registry (ACR)

For SPNs, the ACR uses the same case-finding approach as for first primary neoplasms [28]. Neoplasms are determined to be SPNs, instead of the recurrence or progression of the first primary neoplasm, by following the guidelines set out by the Surveillance Epidemiology and End Results Program for solid tumors [29] and hematopoietic cancers [30]. For deaths, the ACR undertakes a monthly linkage with the Vital Statistics database [25], with all causes of death being coded using the International Classification of Diseases (ICD), 10th revision [31].

#### 2.4.2. Discharge Abstract Database (DAD)

Available from 1 April 1993, the DAD provides demographic, administrative, and clinical data from acute inpatient care and day surgeries [32]. For each hospitalization, up to 25 diagnoses and 20 procedures are recorded using the ICD, 10th Revision, Canada [33], and the Canadian Classification of Health Interventions, respectively [34], and standardized forms and definitions are provided by the Canadian Institute for Health Information (CIHI). In this study, we used data from 1 April 2002 onwards to avoid inconsistencies due to coding rule changes [35].

#### 2.4.3. National Ambulatory Care Reporting System (NACRS)

The NACRS collects demographic, administrative, clinical, and service-specific data for emergency department visits, day surgeries, urgent care visits, and other ambulatory care or specialty clinic visits [25]. From 1 April 1997 to 31 March 2010, ambulatory care data were captured using Alberta’s provincial reporting system and grouping methodology. However, since 1 April 2010, Alberta submits ambulatory care data to CIHI and utilizes CIHI’s Comprehensive Ambulatory Care Classification System grouping methodology [32]. Due to changes in the coding system over the years, in this study, we only used data from 1 April 2002 onwards.

#### 2.4.4. Practitioner Claims

The Practitioner Claims database contains information about the service utilization of physicians and their patients as well as payment information [32]. These data are collected through claims submitted under an Alberta Health Care Insurance Plan (AHCIP) [25]. Data are coded for billing purposes by physicians or their staff, with up to three diagnoses coded using the ICD, 9th Revision, Clinical Modification [36], and one procedure is coded using the Canadian Classification of Diagnostic, Therapeutic, and Surgical Procedures. For the purposes of this project, data were available from 1 January 2001 onwards.

#### 2.4.5. Long-Term Survivor Questionnaire (LTSQ)

The LTSQ was adapted from the After Completion of Therapy Clinic at St. Jude Children’s Research Hospital and included the standardized Patient Reported Outcomes Measurement Information System (PROMIS) questionnaires. The LTSQ asks about general demographic information such as gender identity, marital status, employment, living arrangements, and school history. In addition, it assesses current health status, including current health problems, medical information, interval history, medications and supplements, reproductive health, sexual health, and sun sensitivity. The LTSQ also asks about various health behaviors (e.g., physical activity, smoking/drug/cannabis use). Finally, the PROMIS measures included in the LTSQ were used to ask questions pertaining to depression, anxiety, social relationships, pain interference, and sleep. Different forms of the LTSQ exist to capture parent-proxy (for survivors 5–17 years of age) and self-report (for survivors ≥11 years of age) data [37]. The Alberta Children’s Hospital LTSC has administered the LTSQ since 2010. The questionnaire has been revised since its inception, with the most recent version being implemented electronically in 2020.

## 3. Results

### 3.1. Cohort Characteristics

Table 1 and Figure 1 present the descriptive characteristics of the cohort overall and for 5-year survivors (i.e., children who survived at least five years from their date of cancer diagnosis). Overall, a total of 2581 children were diagnosed with a first primary neoplasm before the age of 18 years in Alberta, Canada between 1 January 2000 and 31 December 2018, 1385 (53.7%) of whom were 5-year survivors. Approximately 48% of the cohort was female, with approximately 36% of cases occurring in the Calgary zone and 32% of cases occurring in the Edmonton zone. Nearly 40% of the cohort was diagnosed between 0 and 4 years of age, and the most frequent diagnoses were leukemias, myeloproliferative diseases, and myelodysplastic diseases (25.3%), central nervous system (CNS) and miscellaneous intracranial and intraspinal neoplasms (24.2%), and lymphomas and reticuloendothelial neoplasms (14.9%). Descriptive characteristics were generally similar when restricted to 5-year survivors.

For individuals with detailed treatment information available (*n* = 1745; 67.6%), nearly all received radiotherapy, chemotherapy, and/or surgery. Chemotherapy only was the most common form of treatment, followed by chemotherapy and surgery, and chemotherapy, radiotherapy, and surgery. Detailed descriptive statistics for chemotherapy agents and radiotherapy sites are shown in Table 2 and Appendix A. The most common chemotherapy agents were Vincristine and Cyclophosphamide, with a median dose of 3002.6 mg/m^2^ observed for the latter. In total, 525 individuals received radiotherapy, and doses were available for 513 (97.7%) individuals. The median dose was 2550 cGy, and the most frequently irradiated sites were the CNS (*n* = 266), abdomen (*n* = 86), and sites other than those specified (*n* = 85). Compared with the overall cohort, individuals for whom detailed treatment information was not available were more likely to be diagnosed with a central nervous system tumor (29.6% vs. 24.2%), be older at diagnosis (median 15.4 years vs. 7.6 years), and not be residing in Calgary (34.1% vs. 35.5%) or Edmonton (29.9% vs. 31.6%) zones (Table 3). Manual treatment abstraction is ongoing for all those individuals missing detailed treatment information.

By the study exit date (31 December 2018), the median attained age was 15.1 years (IQR: 8.7–20.8), and the median follow-up time from cancer diagnosis was 5.6 years (IQR: 1.9–10.9) (Table 1). As expected, the median attained age and follow-up time increased when restricted to 5-year survivors to 19.7 (IQR: 14.0–24.5) and 10.3 (IQR: 7.5–13.7), respectively.

### 3.2. Late Effects

At the time of publication, data linkages have been undertaken with all population-based registries and administrative health databases. Work is ongoing to abstract late effects’ data from the LTSQs, which were paper-based until 2020. The results are thus only presented for outcomes where data collection is complete.

Figure 2 shows a flowchart of all data linkages. SPNs and vital status information were ascertainable for the entire cohort (Table 4). A total of 94 SPNs were observed among 83 survivors of childhood cancer, 55 of which were observed after 5-year survivorship in 48 survivors. In terms of mortality, 408 (15.8%) of the survivors of childhood cancer included in the overall cohort died by the study exit date, though the proportion substantially decreased when restricted to 5-year survivors, at 2.7%. For the survivors of childhood cancer that were able to be linked to the administrative health databases (i.e., not missing a ULI (*n* = 1) and who were not dead or lost to follow-up before the beginning of each database), the following numbers were observed: 2254 (87.3%) appeared in DAD, with a total of 16,669 records discharged; 2533 (98.1%) appeared in NACRS, with 445,150 records ambulatory/outpatient events observed; and 2549 (98.8%) appeared in Practitioner Claims, equating to 396,074 records. The corresponding number of survivors and observed records for each database for 5-year survivors can be found in Table 4. It is worthwhile to note that some events from these databases may be related to care or side effects occurring during the treatment phase, particularly in regard to events before 5-year survivorship, and that not all healthcare utilization will be related to their past cancer diagnosis.

## 4. Discussion

### 4.1. Strengths of the Research Program

The Alberta Childhood Cancer Survivorship Research Program will comprehensively assess the risk of cancer-related adverse outcomes among contemporarily diagnosed and treated survivors of childhood cancer. Ultimately, our findings will inform clinicians and policymakers on long-term health risks and the importance of lifelong care for survivors of childhood cancer and will assist patients through their utilization in counseling, educating, and empowering survivors. The main strength of this program is that it combines the population-based nature of cancer registry-based cohorts with detailed treatment exposure data, which makes our findings both generalizable and clinically relevant. Additional strengths include that the cohort is based upon the ACR, which has achieved the highest standard of quality for case finding and reporting [26], and that the majority of our outcomes are ascertained through deterministic linkages with population-based databases that allow the comprehensive ascertainment of outcomes and healthcare utilization and avoids nonresponse and recall bias associated with self-report questionnaires. Furthermore, the prioritization of patient-reported outcomes will allow us to critically assess symptom burden as reported by the survivors themselves. Finally, as Alberta utilizes a one-payer, universal healthcare system, the short and late effects reported from this cohort are representative of the true burden of late effects in this cancer survivor population, unlike in other settings where outcomes and healthcare utilization may be impacted by the availability of insurance.

### 4.2. Comparisons with the Literature

There are several childhood cancer survivor cohorts worldwide; major studies include the North American Childhood Cancer Survivor Study (CCSS) in the USA and Canada, the Adult Life after Childhood Cancer in Scandinavia study (ALiCCS), the British Childhood Cancer Survivor Study (BCCSS), the Dutch Childhood Oncology Group (DCOG) LATER study, the Swiss Childhood Cancer Survivor Study (SCCSS), and the French Childhood Cancer Survivor Study (FCCSS) [18,38,39,40]. While the CCSS is a multi-institution study and thus not population-based, all the previously mentioned European cohorts are registry-based and thus are representative of all children treated in those countries, although the FCCSS does not cover the entire French population. The strength of the CCSS, however, is that it includes detailed treatment exposure and follow-up information for all participants, an essential for informing clinical recommendations. Our program is unique in that it is both population-based and clinical in nature and thus fills a current void in the literature. Other differences between these cohorts include their inclusion criteria for age, diagnosis years, cancer types, and entry for follow-up, as shown in Table 5. While our program is the smallest of the studies, this relates to the fact that we included substantially fewer diagnosis years and that we included only one province of Canada rather than the entire country. Our cohort is also the only one that currently includes diagnoses after 2008 with the most contemporary therapies, most of which have yet to be explored in relation to late effects. Finally, our program begins follow-up at diagnosis, which corresponds with the definition of a survivor established by the National Cancer Institute (NCI). Using this inclusive definition, we will be able to more comprehensively understand survivorship needs among this young population. It also allows us the flexibility to define the observation period based on the research question being asked. For example, it would be most appropriate to start late-effect studies at the date of treatment completion, as recommended by the NCI, because late effects can occur before 5-year survivorship; for this example, our study will overcome limitations in the literature by not missing these early adverse events, thus providing a more accurate understanding of late effects among survivors of childhood cancer.

In addition to the CCSS, which includes one Canadian center (i.e., The Hospital for Sick Children, Toronto), there are several ongoing initiatives in Canada related to childhood cancer survivorship. The first, the Childhood, Adolescent, and Young Adult Cancer Survivors Research Program (CAYACS), includes survivors of cancer diagnosed under the age of 25 years in the province of British Columbia [41]. Although this cohort includes historical cases compared with our cohort, it is the most comparable in terms of cohort constitution and ascertainment of outcomes through deterministic data linkage. The second initiative is the Pediatric Oncology Group of Ontario Networked Information System (POGONIS), which was established in 1985. While not a cohort per se, cohorts can be pulled from the POGONIS database that contains detailed clinical information and specifics about each child’s diagnosis, treatment, complications, and long-term outcomes. These data have thus led to numerous survivorship-related publications [42,43]. Similarly, while we use the Alberta portion of CYP-C, the national CYP-C database includes survivors of childhood cancer from all 17 pediatric hospitals in Canada and can be used to generate cohorts for survivorship research. As we utilize CYP-C as the foundation for our clinical data, our cohort is comparable to POGONIS and the national CYP-C data for these variables. However, our study has the added strength of being population-based, ensuring that our results are representative of childhood cancer survivorship care and needs in Alberta.

### 4.3. Limitations

While the contemporary nature of the foundational cohort of the Alberta Childhood Cancer Survivorship Research Program sets it apart from existing studies, it is simultaneously a limitation, as the cohort needs additional time to mature in order to observe a sufficient number of adverse outcomes so that they can be investigated with increased granularity. Furthermore, while we do include detailed treatment exposure data, these data are primarily sourced through linkage with the CYP-C program, and thus we are restrained to the level of detail that was determined at the time of data collection. As a result, not all chemotherapy agents have cumulative doses, and radiotherapy sites may not be as precise as needed for some hypotheses.

### 4.4. Future Directions

The immediate next steps for the program include completing data collection for treatment exposure data and psychosocial outcomes and investigating the risk of mortality, subsequent cancers, and cardiovascular disease. The foundational cohort will be prospectively expanded every five years to add newly diagnosed survivors of childhood cancer. Similarly, each data linkage will be updated at the same time, which will allow for the continued longitudinal assessment of the baseline cohort, as well as the ascertainment of all exposures and outcomes for newly diagnosed survivors. As the LTSQ is now collected electronically, these data will also be prospectively pulled at the same time as all other outcome data. Additionally, we hope to explore healthcare utilization among survivors of childhood cancer in Alberta and suggest risk stratification models for follow-up care using machine-learning-based predictive modeling. Finally, while we report here on the data that are currently available, it is our intention to grow the Alberta Childhood Cancer Survivorship Program to include other databases and samples; for example, we are particularly interested in linking with the Pharmaceutical Information Network [25], which records current and previous medication use, as well as pharmacogenetic information and biobanking samples that are already being collected through other initiatives in Alberta.

## 5. Conclusions

The Alberta Childhood Cancer Survivorship Research Program is an important cohort for understanding long-term survival and adverse outcomes in Albertan survivors of childhood cancer. The results from the program will add to the literature and may inform clinical care and tertiary prevention strategies seeking to improve care and reduce cancer-related sequelae in this survivor population.

## Figures and Tables

**Figure 1 cancers-15-03932-f001:**
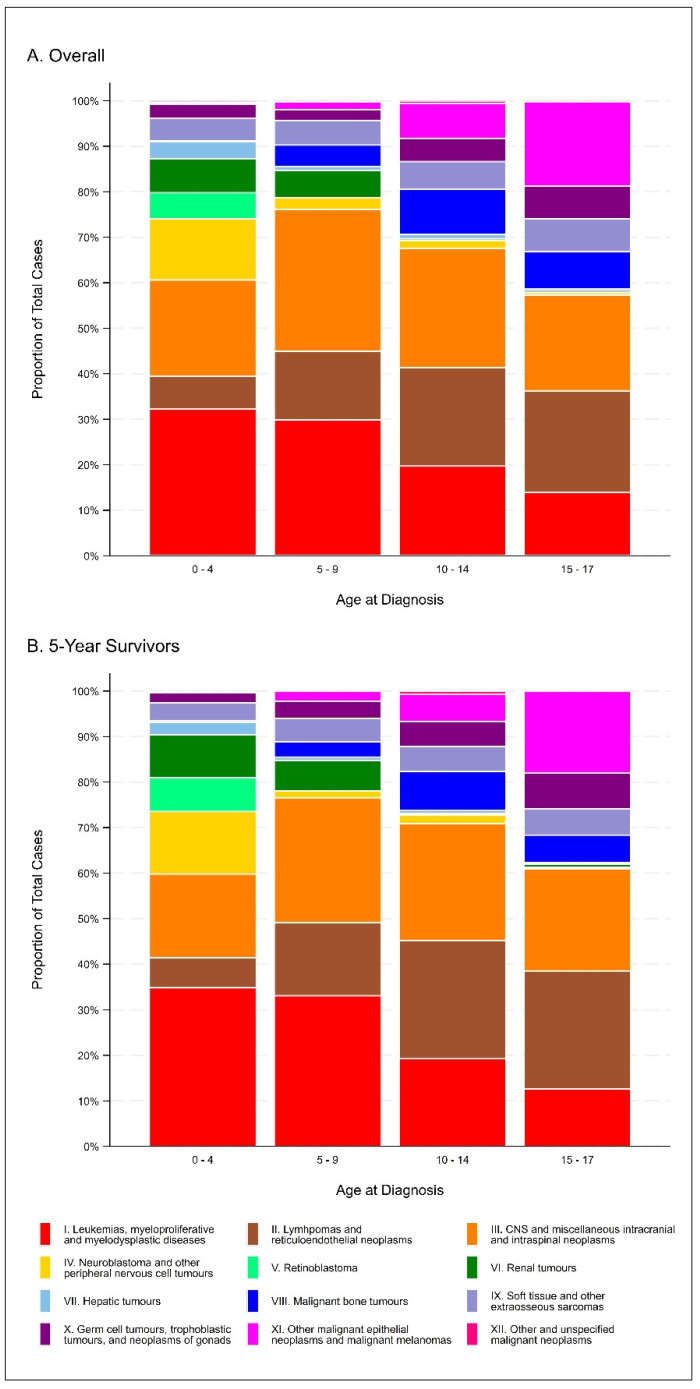
Distribution of ICCC-3 diagnosis categories, overall (**A**) and among 5-year survivors (**B**).

**Figure 2 cancers-15-03932-f002:**
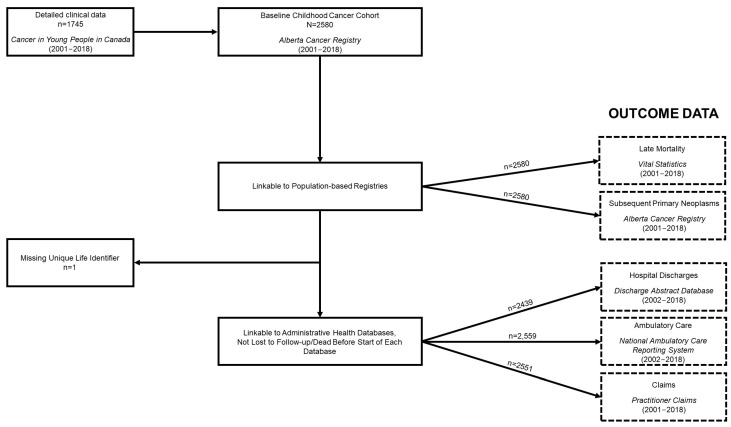
Alberta Childhood Cancer Survivorship Research Program cohort constitution flowchart.

**Table 1 cancers-15-03932-t001:** Cohort characteristics, overall and among 5-year survivors.

Characteristic	Overall Number (%)	5-Year Survivors Number (%)
Total	2581 (100.0)	1385 (100.0)
Sex		
Male	1354 (52.5)	740 (53.4)
Female	1227 (47.5)	645 (46.6)
Time Since Diagnosis (years) ^1^		
0–<5	1196 (46.3)	0 (0.0)
5–<10	627 (24.3)	627 (45.3)
10–<14	522 (20.2)	522 (37.7)
≥ 15	236 (9.1)	236 (17.0)
Median (IQR)	5.6 (1.9–10.9)	10.3 (7.5–13.7)
ICCC-3 Diagnosis Category		
Leukemias, myeloproliferative diseases, and myelodysplastic diseases	654 (25.3)	364 (26.3)
Lymphomas and reticuloendothelial neoplasms	385 (14.9)	233 (16.8)
CNS and miscellaneous intracranial and intraspinal neoplasms	624 (24.2)	314 (22.7)
Neuroblastoma and other peripheral nervous cell tumors	161 (6.2)	81 (5.9)
Retinoblastoma	57 (2.2)	38 (2.7)
Renal tumors	109 (4.2)	69 (5.0)
Hepatic tumors	48 (1.9)	19 (1.4)
Malignant bone tumors	124 (4.8)	56 (4.0)
Soft tissue and other extraosseous sarcomas	148 (5.7)	68 (4.9)
Germ cell tumors, trophoblastic tumors, and neoplasms of gonads	109 (4.2)	61 (4.4)
Other malignant epithelial neoplasms and malignant melanomas	153 (5.9)	78 (5.6)
Other and unspecified malignant neoplasms	9 (0.4)	4 (0.3)
Year of Diagnosis		
2001–2005	620 (24.0)	502 (36.3)
2006–2010	676 (26.2)	549 (39.6)
2011–2015	786 (30.5)	334 (24.1)
2016–2018	499 (19.3)	0 (0.0)
Age at Diagnosis (years)		
0–4	1000 (38.7)	510 (36.8)
5–9	503 (19.5)	269 (19.4)
10–14	554 (21.5)	312 (22.5)
15–17	524 (20.3)	294 (21.2)
Median (IQR)	7.6 (2.9–14.1)	8.1 (3.2–14.4)
Attained Age (years) ^1^		
0–4	318 (12.3)	0 (0.0)
5–9	451 (17.5)	134 (9.7)
10–14	510 (19.8)	279 (20.1)
15–19	582 (22.6)	302 (21.8)
20–24	404 (15.7)	354 (25.6)
25–29	226 (8.8)	226 (16.3)
30–34	89 (3.5)	89 (6.4)
≥ 35	1 (<0.1)	1 (0.1)
Median (IQR)	15.1 (8.7–20.8)	19.7 (14.0–24.5)
CYP-C Treatment		
No treatment	70 (2.7)	19 (1.4)
CT only	588 (22.8)	317 (22.9)
RT only	10 (0.4)	2 (0.1)
Surgery only	215 (8.3)	127 (9.2)
CT + RT	154 (6.0)	81 (5.9)
CT + Surgery	347 (13.4)	178 (12.9)
RT + Surgery	71 (2.8)	39 (2.8)
CT + RT + Surgery	290 (11.2)	140 (10.1)
Patients with missing treatment information	836 (32.4)	482 (34.8)
Zone of Diagnosis		
South	164 (6.4)	89 (6.4)
Calgary	915 (35.5)	471 (34.0)
Central	343 (13.3)	202 (14.6)
Edmonton	815 (31.6)	440 (31.8)
North	343 (13.3)	182 (13.1)
Alberta, zone unknown	1 (<0.1)	1 (0.1)

^1^ Censored to date of death, last known date (i.e., loss to follow-up), or end of study (i.e., 31 December 2018), whichever occurs first.

**Table 2 cancers-15-03932-t002:** Available chemotherapy treatment agents and radiotherapy with cumulative doses.

Chemotherapy Agent	Number of Survivors	Dose Available (%)	Median (mg/m^2^) (IQR)
Bleomycin Blenoxane Bleo	96	95 (99.0)	60.6 (46.0–69.1)
Busulphan Busulfan (Myleran)	4	4 (100.0)	346.3 (295.3–429.1)
Carboplatin CBDCA Paraplatin Carboplatinum	209	164 (78.5)	1480.5 (983.0–2250.2)
Carmustine (BCNU) Bis-Chloroethyl-Nitrosourea BiCNU	2	2 (100.0)	309.9 (58.4–561.4)
Cisplatin CDDP Platinol Cisplatinum Cis-diamminedicloro-platinum II P	232	232 (100.0)	368.4 (230.4–454.3)
Cyclophosphamide Cytoxan CTX Procytox	837	831 (99.3)	3002.6 (1124.9–5293.9)
Cytarabine (IT ONLY) Ara-C Cytosar Cytosine arabinoside	558	441 (79.0)	95.6 (75.4–188.3)
Cytarabine (ONLY IV ≥ 500 mg/m^2^ per dose) Ara-C Cytosar Cytosine arabinoside	146	145 (99.3)	15,009.4 (10,349.3–24,129.0)
Daunomycin Daunorubicin Cerubidine DNR	300	299 (99.7)	102.1 (96.5–166.2)
Doxorubicin Adriamycin ADR	786	780 (99.2)	109.9 (74.7–217.8)
Doxorubicin-Pegylated Liposomal (DOXIL) PLD	4	4 (100.0)	111.9 (78.0–436.7)
Etoposide (VP16) VePesid ETOP	564	559 (99.1)	1335.1 (598.9–1875.0)
Etoposide Phosphate	3	2 (66.7)	2107.5 (1200.0–3015.0)
Hydrocortisone (IT ONLY)	113	89 (78.8)	80.3 (40.8–162.6)
Idarubicin Idamycin 4-Demethoxydaunorubicin	28	28 (100.0)	11.0 (09.9–20.9)
Ifosfamide Isophosphamide IFOS Ifex Holoxan	217	215 (99.1)	20,354.5 (6052.1–46,236.1)
Lomustine (CCNU) CeeNU Chloroethyl-Cyclohexyl-Nitrosurea	20	18 (90.0)	445.8 (299.3–568.5)
Melphalan L-PAM Alkeran L-Sarcolysin	5	5 (100.0)	177.5 (100.4–192.5)
Methotrexate (IT ONLY) MTX Amethopterin	558	446 (79.9)	243.6 (113.1–300.0)
Methotrexate (IV ≥ 500 mg/m^2^ ONLY) MTX Amethopterin	355	352 (99.2)	14,491.6 (6319.5–20,000.0)
Mitoxantrone Novantrone DHAD Dihydrochloride	57	57 (100.0)	46.6 (34.4–48.5)
Oxaliplatin Eloxatin	4	4 (100.0)	178.6 (53.5–286.2)
Procarbazine PCB Natulan Matulane	7	6 (85.7)	733.0 (62.3–1600.0)
Teniposide (Vumon) VM-26	21	21 (100.0)	404.2 (383.7–529.0)
Thiotepa TESPA Triethylene Thiophosphoramide	12	11 (91.7)	502.2 (16.8–1271.8)
**Radiotherapy Site**	**Patients**	**Patients** **with** **Dosage** **(%)**	**Median (cGy)** **(IQR)**
All Sites Combined	525	513 (97.7)	2550 (1800–4776)
Abdomen	86	86 (100.0)	1490 (1080–2400)
Central Nervous System ^1^	266	265 (99.6)	3600 (1800–5400)
Chest	6	6 (100.0)	2325 (1500–4500)
Face ^2^	16	16 (100.0)	3870 (2000–4320)
Limb	43	42 (97.7)	4500 (2200–5000)
Liver	3	3 (100.0)	600 (450–1200)
Lung	32	32 (100.0)	1500 (1200–2130)
Lymph Nodes ^3^	56	56 (100.0)	2100.0 (2100–3075)
Nasopharynx	5	5 (100.0)	4150 (3600–4500)
Neck	56	56 (100.0)	2100 (2100–3375)
Pelvis	35	35 (100.0)	2100 (1050–3600)
Skull	18	17 (94.4)	2400 (1500–4500)
Spleen	11	11 (100.0)	2100 (2100–2100)
Testis	3	3 (100.0)	3600 (1200–8659)
Thorax ^4^	40	40 (100.0)	2100 (2100–2550)
Other	85	82 (96.5)	2603 (2000–5000)
Missing ^5^	17	3 (17.6)	2160 (1440–5400)

^1^ Category consists of site codes “Brain,” “Spine”, and “Craniospinal”. ^2^ Category consists of site codes “Face”, “Orbit”, and “Parotid”. ^3^ Category consists of site codes “Lymph Nodes” and “Mantle Nodes”. ^4^ Category consists of site codes “Mediastinum” and “Thorax”. ^5^ Category consists of site code “Not available” and records with missing site code value. Presented data are based on treatment information available as of 4 May 2022 (for Alberta Children’s Hospital) and 30 November 2022 (for Stollery Children’s Hospital).

**Table 3 cancers-15-03932-t003:** Characteristics of cohort participants who are not in the Cancer in Young People in Canada consortium.

Characteristic	Overall Number (%)	5-Year Survivors Number (%)
Total	836 (100.0)	482 (100.0)
Sex		
Male	399 (47.7)	238 (49.4)
Female	437 (52.3)	244 (50.6)
Time Since Diagnosis (years) ^1^		
0–<5	354 (42.3)	0 (0.0)
5–<10	243 (29.1)	243 (50.4)
10–<14	174 (20.8)	174 (36.1)
≥ 15	65 (7.8)	65 (13.5)
Median (IQR)	6.0 (2.2–10.8)	9.9 (7.1–13.2)
ICCC-3 Diagnosis Category		
Leukemias, myeloproliferative diseases, and myelodysplastic diseases	108 (12.9)	53 (11.0)
Lymphomas and reticuloendothelial neoplasms	144 (17.2)	99 (20.5)
CNS and miscellaneous intracranial and intraspinal neoplasms	247 (29.6)	147 (30.5)
Neuroblastoma and other peripheral nervous cell tumors	16 (1.9)	7 (1.5)
Retinoblastoma	40 (4.8)	31 (6.4)
Renal tumors	6 (0.7)	5 (1.0)
Hepatic tumors	3 (0.4)	1 (0.2)
Malignant bone tumors	45 (5.4)	19 (3.9)
Soft tissue and other extraosseous sarcomas	39 (4.7)	21 (4.4)
Germ cell tumors, trophoblastic tumors, and neoplasms of gonads	55 (6.6)	31 (6.4)
Other malignant epithelial neoplasms and malignant melanomas	131 (15.7)	67 (13.9)
Other and unspecified malignant neoplasms	2 (0.2)	1 (0.2)
Year of Diagnosis		
2001–2005	215 (25.7)	165 (34.2)
2006–2010	218 (26.1)	179 (37.1)
2011–2015	276 (33.0)	138 (28.6)
2016–2018	127 (15.2)	0 (0.0)
Age at Diagnosis (years)		
0–4	161 (19.3)	80 (16.6)
5–9	65 (7.8)	32 (6.6)
10–14	128 (15.3)	78 (16.2)
15–17	482 (57.7)	292 (60.6)
Median (IQR)	15.4 (8.5–16.8)	15.6 (11.0–17.0)
Attained Age (years) ^1^		
0–4	63 (7.5)	0 (0.0)
5–9	73 (8.7)	31 (6.4)
10–14	75 (9.0)	37 (7.7)
15–19	211 (25.2)	48 (10.0)
20–24	201 (24.0)	153 (31.7)
25–29	138 (16.5)	138 (28.6)
30–34	74 (8.9)	74 (15.4)
≥ 35	1 (0.1)	1 (0.2)
Median (IQR)	19.9 (14.8–25.2)	24.2 (20.4–28.2)
Zone of Diagnosis		
South	64 (7.7)	38 (7.9)
Calgary	285 (34.1)	151 (31.3)
Central	118 (14.1)	77 (16.0)
Edmonton	250 (29.9)	152 (31.5)
North	118 (14.1)	63 (13.1)
Alberta, zone unknown	1 (0.1)	1 (0.2)

^1^ Censored to date of death, last known date (i.e., loss to follow-up), or end of study (i.e., 31 December 2018), whichever occurs first.

**Table 4 cancers-15-03932-t004:** Late effects data, overall and after 5-year survivorship.

Database and Outcome	Overall (%)	5-Year Survivors (%)
Alberta Cancer Registry		
Subsequent primary neoplasms observed	94	55
Number of survivors	83 (3.2)	48 (3.5)
Vital Statistics		
Alive	2173 (84.2)	1347 (97.3)
Deceased	408 (15.8)	38 (2.7)
Discharge Abstract Database		
Number of inpatient events	16,669	995
Number of survivors	2254 (87.3)	331 (23.9)
National Ambulatory Care Reporting System		
Number of ambulatory/outpatient events	445,150	60,906
Number of survivors	2533 (98.1)	1296 (93.6)
Practitioner Claims		
Number of claims records	396,074	88,885
Number of survivors	2549 (98.8)	1317 (95.1)

**Table 5 cancers-15-03932-t005:** Characteristics of childhood cancer survivorship cohorts.

**Cohort**	Alberta Childhood Cancer Survivorship Research Program	Childhood, Adolescent, and Young Adult Cancer Survivors Research Program	North American Childhood Cancer Survivor Study	Adult Life after Childhood Cancer in Scandinavia	British Childhood Cancer Survivor Study	Dutch Childhood Oncology Group LATER	Swiss Childhood Cancer Survivor Study	French Childhood Cancer Survivor Study
**Country**	Canada	Canada	Canada United States	Denmark Finland Iceland Norway Sweden	England Scotland Wales	The Netherlands	Switzerland	France
**Identification of survivors**	Registry-based	Registry-based	Institution-based	Registry-based	Registry-based	Registry-based	Registry-based	Registry-based
**Number of survivors**	2580	3841	38,036	33,160	34,490	6168	4405	7670
**Age at cancer diagnosis**	0–17	0–24	0–20	0–19	0–14	0–17	0–20	0–19
**Diagnosis years**	2001–2018	1970–1995	1970–1999	1943–2008	1940–2006	1963–2002	1976–2014	1945–2000
**Cancer types**	All	All	Leukemia, tumor of the central nervous system, Hodgkin lymphoma, non-Hodgkin lymphoma, Wilms tumor, neuroblastoma, soft-tissue sarcoma, and bone tumor	All	All	All	All	All except leukemia
**Entry for follow-up**	Diagnosis	5 years post-diagnosis	5 years post-diagnosis	1-year post-diagnosis	5 years post-diagnosis	5 years post-diagnosis	5 years post-diagnosis	5 years post-diagnosis
**Detailed treatment information**	Yes (collection ongoing)	No	Yes	No	No	No	No	No

## Data Availability

The data underlying this article cannot be shared publicly, as they contain semi-identifiable information that could compromise research participant privacy. However, additional summary tables of count data are available from the corresponding author upon request.

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
