# Peer review of "Alberta Childhood Cancer Survivorship Research Program"

_cancers, 2023, doi:10.3390/cancers15153932_

Round 1
Reviewer 1 Report
This is a well-written manuscript describing the Alberta Childhood Cancer Survivorship Research Program, where data from all childhood cancer survivors within the province of Alberta, Canada have been collected for all patients with a diagnosis between 2001 to 2018. This report details the database itself, its linkage with other health records in the setting of a single-payer universal health care system. The database and data presented here represent an opportunity to learn more about late effects of more contemporary cancer treatment regimens given when compared to other published studies. There is a lot of nice information shared within this manuscript, and likely methodologies that could be useful in other single-payer healthcare settings to start a similar registry where they don’t already exist.
The manuscript could be strengthened by a little more of a granular look at the outcomes that are more fully fleshed out at this point, even if not completely detailed to date, including how overall mortality in this cohort compares to other survivorship cohorts; secondary cancer incidence, etc? Cardiovascular events would be great to have more data for also as there are many unanswered questions about late effects of anthracyclines/dosing/use of dexrazoxane. If there are any other findings that could be compared to other survivorship studies, I think it would significantly strengthen this manuscript, even if the authors acknowledge that not all the data are mature. In its current form, this is a more methodologic and descriptive manuscript.
Minor editing suggestions:
Line 35 (abstract), “of which 1379” should be “of whom 1379”
Line 194: “chemotherapy agent was” should be “chemotherapy agents were”
- Also here, I think Vincristine should be added to Table 2 if it is going to be noted here as the most common chemo given. I see it in the supplemental data, but is there a reason it wasn’t included in this table?
Line 195: “radiotherapy, of which doses” should be “radiotherapy, for whom doses”
Under 2.4 Late Effect Ascertainment, the subgroups are numbered 2.3 instead of 2.4
Well written, no concerns about writing/language.
Author Response
Please see attached Word document.

Reviewer 2 Report
This is a detailed study on a survivor programme that links treatment with survivorship.
It is nicely written and presented.
I suggest that the program's modifiable and non-modifiable aspects be described and examined. This would increase its utility and applicability.
"4.1 Summary" is used as a subheading in the discussion. This, I believe, is inappropriate. Please select a more appropriate subheading. Perhaps 'benefits of program' or 'strengths of program'.
Author Response
Please see attached Word document.

Reviewer 3 Report
The Authors present an interesting manuscript: " Alberta Childhood Cancer Survivorship Research Program" where all the different parts are well reported and well divided . Aim of the paper is clear. I wonder only minor points:
1. When you describe in the paragraph 3.2 Late effects and report the terms of mortality, 437 (16.9%) : for what they died? this information could be interesting as well as in
2. which type of late effects did you observe in your analysis? I don't ask for a deep analysis
3. the same request should be addressed for the psychosocial aspects of long term survivors. A general look.
4. Conclusions are promising but if you claim this conclusion you need to specify something more for being able to modify the provided treatment for cancer
Author Response
Please see attached Word document.

Round 2
Reviewer 1 Report
Thank you for your edits/additions based on the reviews provided. As a methods type manuscript this is very nicely laid out and reads well. No further edits requested.